# Insights on Aggregation of Hen Egg-White Lysozyme from Raman Spectroscopy and MD Simulations

**DOI:** 10.3390/molecules27207122

**Published:** 2022-10-21

**Authors:** Divya Chalapathi, Amrendra Kumar, Pratik Behera, Shijulal Nelson Sathi, Rajaram Swaminathan, Chandrabhas Narayana

**Affiliations:** 1Chemistry and Physics of Materials Unit, School of Advanced Materials, Jawaharlal Nehru Centre for Advanced Scientific Research, Bengaluru 560064, India; 2Department of Bioscience and Bioengineering, Indian Institute of Technology-Guwahati, North Amingaon, Guwahati 781039, India; 3Transdisciplinary Biology Program, Rajiv Gandhi Centre for Biotechnology, Thycaud Post, Poojapura, Thiruvananthapuram 695014, India

**Keywords:** protein aggregation, Raman spectroscopy, molecular dynamics, protein–protein interactions, structural biology

## Abstract

Protein misfolding and aggregation play a significant role in several neurodegenerative diseases. In the present work, the spontaneous aggregation of hen egg-white lysozyme (HEWL) in an alkaline pH 12.2 at an ambient temperature was studied to obtain molecular insights. The time-dependent changes in spectral peaks indicated the formation of β sheets and their effects on the backbone and amino acids during the aggregation process. Introducing iodoacetamide revealed the crucial role of intermolecular disulphide bonds amidst monomers in the aggregation process. These findings were corroborated by Molecular Dynamics (MD) simulations and protein-docking studies. MD simulations helped establish and visualize the unfolding of the proteins when exposed to an alkaline pH. Protein docking revealed a preferential dimer formation between the HEWL monomers at pH 12.2 compared with the neutral pH. The combination of Raman spectroscopy and MD simulations is a powerful tool to study protein aggregation mechanisms.

## 1. Introduction

As we age, we are more likely to suffer from neurodegenerative diseases such as Alzheimer’s and Parkinson’s [1], which result from the deposition of misfolded proteins and their aggregates. Sometimes genetic mutations, the functioning of a ribosome, or just the occurrence of a random event can alter the folding pathway of proteins upon their production [2]. These toxic protein conformations can sometimes interact with the existing copies of the native protein and alter their state, and hence are “infectious” as shown in the case of prions [3]

Protein aggregation can occur through several unique mechanisms/pathways such as the reversible association of the native monomer, aggregation of the conformationally altered monomer, aggregation of the chemically modified product, nucleation-controlled aggregation, and surface-induced aggregation [4]. Partially unfolded intermediates have dangling bonds, and hence exposed regions that are capable of intermolecular interactions [5]. This complete or partial unfolding can be triggered by various physiochemical parameters, including temperature, pressure, pH, agitation, and chemical environment [6]. The primary sequence of the protein and the secondary and tertiary structures play a vital role in the susceptibility of the protein to unfolding. Broadly, protein aggregation pathways can be classified into two standard models: nucleation-elongation polymerization and isodesmic polymerization [7,8]. In the isodesmic model, proteins become dangerously susceptible to aggregation as there is no critical concentration required for the process to begin.

Human lysozyme, which bears close homology to the HEWL used in this work, has been implicated in systemic nonneuropathic amyloidosis [9], where sometimes kilogram quantities of aggregates in the variant form of this protein can be deposited in the liver, kidney, spleen or gastrointestinal tract [9,10,11]. Thus, HEWL, which is easily available, can serve as an excellent model to understand lysozyme amyloidosis in humans. HEWL has been shown to aggregate under a variety of conditions [12]

The pH plays a vital role as it changes the protonation state of the charged amino acids, the C-terminal, and the N-terminal of the protein. The change in the charge distribution around the protein affects the salt bridges and hydrogen bonds, which are essential for a protein’s strength and overall structure [13]. In an acidic pH environment, the negative charge residues become neutralized, while there is an increase in repulsion amongst the polycationic monomers. Thus, to facilitate aggregation in the acidic conditions, we need to provide external stimuli such as increased temperature or agitation as shown in the previous reports on the aggregation of HEWL in acidic pH at higher temperatures [14,15,16,17]. Sophianopoulus and coworkers were the first to demonstrate aggregation of HEWL at alkaline pH [18]. HEWL aggregation at alkaline pH (12.2) can be achieved at ambient temperature as it has a low net charge at this pH [19,20]. This helps in elucidating the effect of pH without any other interferences. This aggregation process is directed by the isodesmic pathway. Hence, it can be studied at relatively dilute concentrations and provide insights into the protein aggregation mechanism.

Various biochemical analysis techniques, including dynamic light scattering (DLS) [21], gel electrophoresis [22], size-exclusion chromatography [23,24,25], atomic force microscopy (AFM) [14,16,20,26], and electron microscopy [27], have been used to investigate the formation and also estimate the size of protein aggregates. To study protein aggregates, other techniques such as circular dichroism [7], fluorescence anisotropy [20], NMR [28], mass spectroscopy [29], IR spectroscopy [30], and Raman spectroscopy [14,15,16,31,32] have also been employed. These techniques provide a picture of the protein aggregates by probing molecular interactions. In particular, Raman spectroscopy is a non-destructive and non-invasive tool that can be used to study a sample in any form, be it a solid, liquid, or gas. Raman spectroscopy is advantageous over other techniques such as IR spectroscopy, having no need for any sample preparation. Various components of a Raman spectrum such as the peak position, peak shift, intensities, and full width at half maximum (FWHM), etc., can reveal information regarding various molecular aspects of a protein and hence provide sensitive and crucial information on the unfolding and aggregation of the protein. A time-dependent Raman study thus reveals a picture of the changes in the various vibrational modes as a function of time, hence correlating to the real-time changes. Under diverse conditions, sometimes variations of the standard Raman spectroscopy such as Resonant Raman, Surface-Enhanced Raman, and FT-Raman have been used to obtain specific information about biomolecular behaviors without becoming lost amidst the vast information that exists in a Raman spectrum [32]. The analysis of a Raman spectrum can reveal a variety of information that otherwise require a range of experiments, thus making it a versatile tool.

Interpreting the various peaks of a Raman spectrum can provide insights into the changes in specific regions of the protein over time. All the obtained information can be stitched together to obtain a bigger picture. However, a protein is so vast that it is almost impossible to obtain the time-dependent changes of every single atom under varied environmental parameters. To validate the Raman results and obtain more information that might not be directly available through experimental techniques, utilization of a molecular simulation tool can provide the missing links and fill the gaps in information. Over many years, with the advancement of software tools and computer hardware, MD simulations have achieved significant success in understanding protein dynamics [33,34,35,36]. Supercomputers such as Anton are specifically designed to perform MD simulations for very large macromolecules. They can simulate from nanoseconds to seconds, the time-scale most molecular activities take [37]. Despite all these advancements in the field, this technique remains underutilized for protein aggregation, with few MD investigations so far. The studies that have been performed mainly concentrate on amyloid formation using coarse-grained protein models simulated with or without explicit solvents [38]. These methods provide an appropriate picture of how proteins aggregate, but by doing so, the molecular level details are sacrificed. Simulations can provide an extraordinary amount of detail concerning individual particle motions as a function of time and answer specific questions about the properties of a model system more often than experiments on an actual system [39]. 

With such great features, MD simulations do have some limitations, including that they cannot straightforwardly mimic real-life conditions, such as the real time required for a protein to unfold and aggregate and the breaking or formation of new bonds during the simulation run. Similarly, while Raman spectroscopy can reveal molecular insights, certain vibrational modes might be Raman inactive or weak scattering, thus missing the complete picture. Hence, in our current work, for the first time, Raman spectroscopy combined with Molecular Dynamics (MD) simulation was utilized to gain molecular insights into the aggregation process of HEWL at pH 12.2 at ambient temperature. The Raman spectra of the HEWL protein incubated at room temperature in pH 7.0 as a control and in pH 12.2 as the experimental system were investigated at various time points to understand, at a molecular level, their response to an altered pH environment over time. Simultaneously, MD simulations were employed to visualize and understand protein unfolding and the subsequent dimerization through docking studies.

## 2. Results

The UV-Visible absorption spectrum of the HEWL stock was collected (Appendix A), and the absorption value at 280 nm was used to calculate its concentration using an extinction coefficient of 37970 M^−1^cm^−1^ [30]. The absence of any peak in the spectral region 320–350 nm confirms the absence of any aggregation states or UV-absorbing prosthetic groups in the system [40]. The samples were drop-casted onto a hydrophobic glass slide and dried in a dessicator (Figure 1a). The Raman spectra were collected on the coffee ring edge that was formed on the dried spots as seen in Figure 1b. Raman spectra of the dried buffer spots were collected as references (Appendix A). Raman spectrum was collected for the lyophilized monomeric HEWL directly procured from Sigma Aldrich as a reference (Figure 1c). Raman spectra of 120 μM HEWL in pH 7.0 and pH 12.2 were collected at various time points, including 0, 0.5, 1, 3, 6, 12, 24, 48, 72, 96, 120, and 240 h.

As reported by D. Zhang et al., Drop Coating Deposition Raman (DCDR) spectroscopy can be used to obtain Raman spectra by depositing the sample onto a substrate and letting it dry completely. The obtained spectra had high reproducibility and were identical to those obtained from higher protein concentrations [41,42]. Using these results, we chose to conduct our experiments similarly and obtain DCDR spectra for all the time points without being concerned much about the loss of the native structure that would have been present in the solution form. 

At every time point, 10 μL of the aliquoted sample was put onto a siliconized glass substrate, and then placed inside a desiccator for 10 min to completely dry out the water from the sample. As shown in Figure 1a, the drop-casted and dried protein sample formed 4–5 mm diameter spots with rings around them. A 10× magnified image (Figure 1b) showed a raised edge of the ring formed due to the coffee ring effect. The Drop Coated Deposition Raman (DCDR) spectra [41] of the samples were collected at room temperature by randomly scanning the edge of the dried ring by focusing the laser beam to an approximate diameter of 1 μm. 

The comparison of the Raman spectra in Figure 1c shows all the peaks appearing in the dried control and sample in comparison to the monomer powder. The close similarity between the Raman profiles (Figure 1c) obtained from lyophilized HEWL (red) and drop-casted sample (green, blue) indicated no/minimal artefacts in the acquired Raman signals from drop-casted samples. Thus, while using this methodology for sampling and signal acquisition, any change in the Raman profiles can directly be correlated to the changes in the chemical identity of the sample under investigation. The Raman bands assigned to various vibrational modes are shown in Appendix A.

### 2.1. Protein Unfolding

The Raman peak around 505 cm^−1^ attributed to the S–S stretching was monitored over ten days at various time intervals. As seen in Figure 2a, this peak remained constant for the protein in pH 7.0, while in pH 12.2 the peak started to reduce at the 12 h time-point and almost completely disappeared by the end of 240 h (Figure 2b). The Raman peak intensities of 505 cm^−1^ were plotted v.s time for the HEWL in both pH 7.0 and pH 12.2, which provided a clearer understanding of its behaviour (Figure 2c). During the course of aggregation, fewer new disulphide bonds formed between the monomers, strengthening the aggregate. However, the molar ratios of newly formed disulphide bonds among the protein aggregates were almost negligible as compared with molar ratios of disulphide bonds present in natively folded HEWL molecules. This led to a profound decrease in peak intensity at 505 cm^−1^, specifically at 120 and 240 h (Figure 2b) when the aggregates were mature and stable.

Trying to correlate events in real-time changes with MD simulations is quite challenging and takes absurd simulation times. HEWL is a widely studied protein, and the temperature-dependent simulation has been proven to induce unfolding more rapidly without affecting the unfolding pathway [43]. Hence, instead of running an extensive simulation, the protein was simulated at pH 7.0 and pH 12.2 at various temperatures to hasten the unfolding process.

For simulating the effect of pH, two systems using 1HEW crystal structure co-ordinates were created. (a) The protein monomer at neutral pH (pH 7.0) with all its disulphide bonds intact was the control system and was referred to as N, and (b) the protein monomer at high pH (pH 12.2) with its intra-disulphide bonds broken was referred to as HBB. This was carried out to create an environment where the protein is exposed to alkaline pH. When placed at pH 12.2, the protein undergoes an alkaline electrolytic shock, breaking many hydrogen bonds, salt bridges, and disulphide bonds [44], resulting in a loss of secondary and tertiary structure. 

Further, reports show a higher unfolding rate at 498 K [45,46,47]; hence, we subjected the monomers to 450 K for a period of 50 ns each. This threshold of 450 K was defined to induce a restricted unfolding just enough to expose the protein surface for inter-monomeric interactions [48]. The MD simulation trajectories comprising 1000 frames for 50 ns of independent simulation of monomers for structural and dynamic behavior were examined to obtain further insights into the system. The backbone atoms computed for 50 ns indicated that both the monomers retained a stable conformation until 25 ns. In N, there was a slight deviation in trajectory until 4 Å, whereas in HBB, the deviation in the trajectory reached up to 16 Å, confirming that the system in alkaline pH became unstable when the disulphide bonds became broken (Figure 2e). As seen in Figure 2d, throughout the 50 ns of simulation, the secondary structure of N remained mostly intact, while HBB unfolded significantly. On observing the secondary structure of the protein through the simulation, a decrease in α-helical content by about 6% and β-sheet content by 4.4% in the HBB compared with the N was observed (Figure 2f).

### 2.2. Protein Interactions and Aggregation

DLS measurements revealed hydrodynamic radii of the samples, which provides us with a preliminary idea of progress in the aggregation process of the system. On day 10 of the incubation of the HEWL in the buffers, the hydrodynamic radius of the sample in pH 7.0 was around 2 nm (100%), while that of the sample in pH 12.2 was distributed around 10.5 nm (30%) and 30.4 nm (70%) (Appendix A). This demonstrates that in alkaline pH, the protein aggregated, thus resulting in large-sized particles.

The Amide I band of the Raman spectrum of a protein provides information about the secondary structure of the protein [14,21,26,49]. The deconvolution of the Amide I band (Appendix A) revealed its various constituents that included an α helix (around 1657 cm^−1^), β-sheet (around 1671 cm^−1^), β turns (around 1688 cm^−1^), and random coils/unordered protein (around 1647 cm^−1^). As shown in Figure 3a, the position of the Amide I band of the protein in neutral pH remained nearly constant. At the same time, a transformation from an α-helix-rich protein (peak around 1657 cm^−1^) to a β-sheet-rich protein (peak around 1672 cm^−1^) was observed in the alkaline pH over time (Figure 3b). Changes began to appear after a period of 24 h. The Amide I band was deconvoluted, and the percentage secondary structure of the α-helix and β-sheet was plotted as a function of time. It was observed that the α-helical and β-sheet content remained almost similar, averaging around 50% and 23%, respectively, for the protein at pH 7.0 (Figure 3c). However, for the protein at pH 12.2, the α-helical content decreased from about 50% to 20% over ten days (Figure 3d), while the β-sheet content increased from about 23% to about 45%, indicating the uncoiling of α-helices and the formation of new β-sheet structures (Appendix A). The Amide III band around 1200–1300 cm^−1^ also gives insight into the secondary structure of the protein. The decrease in peak intensity around 1280 cm^−1^ correlates to the results obtained from the Amide I band analysis (Figure 3e) by indicating a decreased α-helical content.

We studied the behavior of 120 μM HEWL in pH 12.2 in the presence of iodoacetamide that can react with the free -SH groups to form alkylated HEWL derivatives that prevent the formation of disulphide bonds [50]. Iodoacetamide, an alkylating agent, acts as an inhibitor of aggregation progression by alkylating the free -SH bonds, hence making it unavailable for new intermolecular disulphide bond formation. This causes the retention of the protein’s original conformation. As seen in Figure 3f, there was no shift in the Amide I band of the protein even at 240 h. This confirmed that the reaction of iodoacetamide with free -SH groups in HEWL is indeed inhibiting the protein aggregation by restricting the formation of inter-molecular disulphide bonds. The absence of shift in the Amide I band for HEWL in pH 7.0 in the presence of iodoacetamide, as seen in Appendix A, assures that there was a negligible effect of the iodoacetamide molecule when the disulphide bonds were already intact. Further, this aggregation study was conducted on two concentrations of the HEWL protein, namely 120 μM and 30 μM. From the % β-sheet in the secondary structure content analysis of the Amide I band as seen in Appendix A, it can be concluded that larger aggregates were formed by the 120 μM system compared with the 30 μM system. This is expected, given the isodesmic nature of its aggregation.

Many Raman bands corresponding to various vibrational modes of different organic and inorganic bonds within the molecule contain a wide range of information reflecting their response to their environments. When the 120 μM HEWL protein was incubated in the pH 12.2 buffer, the intensities, shifts in peak positions, and the FWHM of various Raman peaks as a function of time were monitored to reveal more information on the behavior of the corresponding vibrations. The intensity ratio of the peaks at 1341 cm^−1^ and 1363 cm^−1^ serves as a hydrophobicity marker [14,26,49]. As seen in Figure 3g, for HEWL in pH 7.0, the ratio remained constant and low, confirming that the protein retained its structure and remained folded. While in Figure 3h, it can be observed that in pH 12.2 the intensity ratio right from the beginning was high, indicating that the protein had started to unfold, thus exposing the tryptophans and making them hydrophilic. After 24 h, the ratio started to decrease (Figure 3g and Appendix A), indicating that the initially solvent-exposed tryptophans became more buried inside the aggregating protein. 

As seen in Figure 3i, the addition of iodoacetamide prevented the protein from further aggregation, thus causing the Trp to become buried quicker (Figure 3i). In the presence of iodoacetamide, no decrease in the intensity of the 505 cm^−1^ peak (Appendix A) was observed, just as in the control pH 7.0 system. This hints towards the crucial role of new intermolecular disulphide bond formation in promoting large aggregate formation and the stability thereof [51]. 

As shown in Appendix A, the FWHM of the 759 cm^−1^ peak originating from the coupled vibrations of in-phase breathing of benzene and pyrrole in the indole ring increased from about 9 to 12 cm^−1^ as a function of the aggregation time. This indicated a weakening in the coupling due to the distortion of the indole ring [49]. Hence, Trp aligns in different orientations with respect to the initial conformations. This is also supported by a slight increase in the peak position of the Tryptophan–indole ring’s nitrogen around 876 cm^−1^ (Appendix A). The 1446 cm^−1^ peak corresponds to the CH, CH_2_, and CH_3_ deformation and scissoring mode. As seen in Appendix A, the FWHM of this peak increased sharply after the initial alkaline shock, indicating an opening up of the protein due to increasing deformations [14]. As seen in Appendix A, the peak around 830 cm^−1^ corresponding to the Fermi doublet associated with the hydroxyl (OH) of tyrosine residues shifted from 834.5 to 829.7 cm^−1^, indicating an increased hydrogen bonding around that tyrosine molecule [14]. The 900 cm^−1^ is a small peak corresponding to the N–C_α_–C stretching vibration in the skeleton of the α-helical part of the secondary structure. As seen in Appendix A, an increase in the peak position, along with an increase in its FWHM, implies a conformational transformation of the N–C_α_–C skeleton [49]. While the 932 cm^−1^ peak also corresponds to a similar vibrational mode seen in the skeleton, its intensity is proportional to the population of the α-helical structures [14]. As seen in Appendix A, the intensity of this peak decreased over time, indicating a loss in α-helices in the whole protein [49]. By observing the kinetics of formation, we can distinguish amorphous aggregates from amyloid fibrils. The amorphous aggregates formed rapidly without a lag phase, while amyloid fibrils formed with a distinct lag phase in the kinetics [52]. Appendix A represent an immediate change in the parameters under study without the presence of a lag phase, hence indicating the formation of amorphous aggregates of the HEWL protein. 

The protein unfolding was followed by the first step of aggregation, i.e., dimerization. To capture this very first step, the protein evolved from the protein unfolding was docked. A dimer with the largest cluster size was selected, and an MD simulation for 200 ns was conducted. The MD simulation trajectory analysis showed that N had a more stable trajectory than HBB from the root mean square deviation plot, as seen in Appendix A. The last 25 ns of the simulation was very stable in both N and HBB; hence, their corresponding frames were considered for interchain interaction analyses and binding free energy calculation for the protein aggregate. As mentioned, the first step to understanding the aggregation starts by analyzing the interactions between the monomers. Non-bonded interactions (hydrogen bond, salt-bridge, pi-cation, pi–pi) play an essential role in stabilizing the docked structure [53,54,55]. HBB has almost double the number of interactions as in N, which proves a favorable association of the monomers in the case of HBB (Figure 4a).

Analyzing the center-of-mass (COM) distance concerning the simulation time provides insight into the stability of the protein in the dimeric state. From Figure 4b, it is seen that the COM between the monomers of HBB reduced from 27.19 Å to around 23 Å over the simulation duration and stabilized there. A sudden drop from 33.78 Å in the COM distance to around 30 ns was observed in the N system but later remained stable at around 30 Å. The COM distance provides an idea of inter-monomeric adjustments to attain stable configuration and supports the fact that an alkaline pH that disrupts -S–S- bonds and promotes protein aggregation. Binding free energy calculation using the MM-GBSA showed higher affinity among the chains of HBB with the mean binding free energy of −77.53 kcal/mol, whereas the mean binding free energy of N is −62.93 kcal/mol (Appendix A) [56]. We can draw a systematic conclusion about the aggregation processes from the abovementioned studies. In the alkaline pH, the protein readily unfolds with a trigger (temperature), exposing the hydrophobic core to allow for interactions with the solvent. Firstly, we observed that the non-bonded interactions increased on exposure to solvent (water). Secondly, the use of the unfolded states as the initial systems for docking reduced the COM distance in the case of HBB. Throughout the simulation, the monomers were brought together in HBB by a large number of non-bonded interactions. Finally, although the MM-GBSA has limited accuracy and becomes challenging with an unfolded charged protein, it provides us with a primary confirmation that the binding affinity of the monomers is higher in HBB than in N. There are experimental observations that the aggregates form by establishing intermolecular disulphide bonds between monomers. So, we mapped the Cys-Cys distances for all the possible 64 combinations over the 200 ns of the simulation (Appendix A). It was interesting to note that the terminal Cys residues came significantly close (Appendix A), and though they did not come close enough to establish a covalent bond between them, the closeness they developed allowed us to predict the possibility of their involvement in the disulphide bond formation and hence aggregation. In Figure 4c, we can see a visual of the docked structure after 200 ns of the simulation. Here it is visible that there were many inter-monomeric interactions amongst the monomers in HBB, as compared with the very few in N. 

## 3. Discussion

Earlier work has revealed the crucial role played by alkaline pH-induced non-native and intermolecular disulphide bonds in stabilizing the HEWL aggregates [57]. Further work by Ravi and co-workers using force spectroscopy showed that the inhibition of such disulphide bonds can weaken non-covalent interaction forces among the monomers in aggregates [51]. Another study by Ravi et al. provides insights into the isodesmic aggregation of HEWL caused by the formation of crosslinked disulphide bonds [20]. Their steady-state fluorescence, DLS, and AFM measurements confirmed that the HEWL aggregates formed independent of the initial concentration of the monomeric protein, thus proving the process to be isodesmic. FRET was used to establish the formation of new intermolecular contacts, resulting in the formation of aggregate polymers. However, the precise change in the immediate molecular environment of interacting monomers leading to the observed aggregation remained unknown. In this work we attempted to shed light on the nature of molecular interactions leading to dimerization and aggregation of HEWL monomers using MD simulations and Raman analysis.

### 3.1. Protein Unfolding

The Raman experiments suggest that there is a distribution in the S-S bond distances, indicating a tendency of breakage of these bonds. A similar observation was made in a study of HEWL aggregation in an acidic pH environment. This behaviour was assigned to the tertiary realignment of the core protein and/or hydrolysis of the disulphide bonds [14]. This result is also consistent with earlier work which showed the free population of thiols in HEWL increased gradually from 0 to 100 h of incubation in alkaline pH [57]. This confirms that at an alkaline pH, the protein’s disulphide bonds tend to break, resulting in the partial unfolding of the protein.

Overall, the analysis of the 505 cm^−1^ peaks in the Raman spectra and the MD simulations of the monomer in both conditions confirmed that the disulphide bond breakage under alkaline conditions could be one of the major causes of the extensive unfolding of the protein, leading to its higher propensity towards aggregation. 

### 3.2. Protein Interactions and Aggregation

The analyses of the Amide I and Amide III bands suggest that there was a decrease in the alpha helical content and an increase in the beta sheet content. 

Many small molecules such as ligands can selectively bind to the native protein, thus preventing major changes to its original structure when physiochemical changes occur in their environment. In the past, molecules such as DTT [57], tri-*N*-acetylchitotriose [58], and iodoacetamide [51] have been used to demonstrate inhibition of HEWL aggregation at alkaline pH in their presence. Our studies using iodoacetamide revealed that the intermolecular disulphide bonds caused polymerization, and hence protein aggregation. 

From the previous Raman spectral analysis, it is evident that there were new bond formations, changes in the conformations and orientations of certain amino acids, an increase in hydrogen bonds of Tyr, and an overall change in the secondary structure of the protein over 240 h. These bits of information can be stitched together to obtain a bigger picture of the overall protein aggregation process. The information on all the kinds of new bonds that form cannot be just obtained through Raman spectroscopy, and hence obtaining assistance from MD simulations to model the protein aggregation and to gain more insight into the various molecular interactions amidst the monomers is a prudent step. 

We used various modes of analysis to predict changes in the HEWL system that promote aggregation. The secondary structure analysis of the protein monomer at 450 K helped establish that the protein in pH 12.2 unfolded the most due to the de-stability of intramolecular bonds. The partially unfolded structures have exposed amino acids that allow for inter-monomeric interactions. This is reflected in the non-bonded interactions map, which showed almost double the number of interactions in HBB than in N. The inter-monomeric center-of-mass distance plot showed that the HBB monomers became closer over the simulation time, indicating a favorable dimer formation.

Therefore, from both Raman and molecular dynamics studies, on analyzing the protein unfolding and aggregating stages, it was evident that the alpha helices unwound and the beta sheets formed during the aggregation steps. The analysis of the various Raman modes and the MD interaction studies revealed the formation of many bonds between the monomers, thus favouring the formation of aggregates. 

## 4. Materials and Methods

### 4.1. Materials

The Hen Egg-White Lysozyme (HEWL) was purchased from Sigma Aldrich. Milli-Q water (Millipore Ltd., St. Louis, MO, USA) with resistivity at 18.2 MΩ.cm at 25 °C was used for all the experiments. Sodium dihydrogen phosphate (NaH_2_PO_4_), disodium hydrogen phosphate (Na_2_HPO_4_), sodium hydroxide (NaOH), sodium azide, dimethylformamide, and iodoacetamide from Sigma Aldrich were used for preparing buffers and inhibition studies.

### 4.2. Lysozyme Sample Preparation and Incubation

A 50 mM stock of sodium dihydrogen phosphate and disodium hydrogen phosphate were prepared, and using 0.1N NaOH, their pH was adjusted to 7.0 and 12.2 respectively. Then, 0.1% w/v sodium azide was added to these buffer solutions to prevent any microbial growth. HEWL stock (5 mg/mL) was freshly prepared using MilliQ water and its concentration was determined from its UV-Vis absorption spectrum. Afterwards, 1 mL of the samples with the required concentration (120 μM and 30 μM) were prepared by diluting the HEWL stock using 50 mM pH 7.0 buffer for control experiments and 50 mM pH 12.2 buffer for aggregation experiments. The prepared samples were incubated in a 25 °C stackable incubator without any agitation for ten days and aliquots were extracted at different time points. 

### 4.3. Lysozyme Sample Preparation for Aggregation Inhibition Studies

For the aggregation inhibition studies, iodoacetamide was dissolved in dimethylformamide to obtain a 1.5 M stock. Then, 2 μL of this stock was added to the HEWL in pH 12.2 buffer at the 2 h time interval. Later, 2 μL was added similarly at 6, 12, and 24 h time intervals, while the sample continued to incubate at 25 °C in an opaque centrifuge tube sealed with parafilm. At the end of 24 h, a total of 8 μL of the iodoacetamide stock solution was present in the aggregating HEWL sample (1 mL, 120 μM, pH 12.2), resulting in a final iodoacetamide concentration of 12 mM. Similarly, a control inhibition experiment was designed by preparing the sample in a pH 7.0 buffer. 

### 4.4. Raman Spectroscopy

A Raman micro-spectroscopy system (LabRam h Evolution, Horiba, Kyoto, Japan) with an open-electrode CCD detector air-cooled to −60 °C was used to obtain the Raman spectra. Raman scattering, generated by ~6 mW power of a 532 nm laser, was recorded for 120 s per window using an Olympus LMPlanFL 50X objective and passed through an 1800 grooves/mm diffraction grating. The LabSpec6 software was used to record the Raman spectrum for each sample in the spectral range of 300 to 1800 cm^−1^. The liquid sample was drop-casted onto a siliconized glass substrate (Hampton Research, Cat. No. h3-223, Aliso Viejo, CA, USA) and then placed in a desiccator for 1 min to completely dry out the water from the sample. 

### 4.5. Raman Data Analysis: (Baseline Correction, Normalization and Curve Fitting)

Each Raman spectrum was inspected on the LabSpec 6 software, and the cosmic radiation spikes were removed. A user-defined baseline was created and subtracted from the spectra on Origin Pro 2016 (OriginLab Ltd.). The spectra were then normalized with respect to the intensity of the phenylalanine band at 1004 cm^−1^, as it is least affected by conformational changes [21]. All the Raman peaks were fit using a Lorentzian curve. The Amide I band around 1650 cm^−1^ was deconvoluted into components fit by a Lorentz curve. The area under the deconvoluted peaks was calculated to form a direct reference to the percentage of a secondary structure component in the protein at a given time point. The time dependences of the various Raman peak parameters such as peak position, FWHM, and intensity were fit using a sigmoidal curve. 

### 4.6. UV-Vis Absorption Spectroscopy

The prepared HEWL stock was diluted ten times using MilliQ water. The UV-visible absorption spectrum of this solution was obtained using Agilent 8453 UV-Visible spectrometer, Santa Clara, CA, USA, and the absorption value at 280 nm estimated the protein concentration of the stock with the extinction coefficient of 37970 M^−1^cm^−1^ using the following equation [59]: 

Concentration of protein = ((Absorbance at 280 nm × 10^7^)/37,970) μM

### 4.7. DLS

The hydrodynamic diameter of the HEWL aggregates was measured in the Zetasizer Ultra (Malvern Instruments, Malvern, UK) using a disposable plastic cell (DTS0012), with the temperature set to 25 °C.

### 4.8. Molecular Dynamics Simulations

#### 4.8.1. Structure of the Hen Egg-White Lysozyme

The HEWL crystal structure was retrieved from RCSB Protein Data Bank (PDB ID: 1HEW) [60] with a resolution of 1.75 Å. It consists of 129 amino acid residues, and the structure is stabilized by four disulphide bonds (C6–C127, C30–C115, C64–C80, and C76–C94). The protein was prepared using Schrodinger Maestro [61]. The pH of the systems was maintained using the PROPKA tool,, LLC, New York, USA and the disulphide bonds were dissolved using the maestro’s build tool. The systems generated were energy minimized using OPLS3e [62,63] to ensure the system did not have any steric clashes or incorrect geometry.

#### 4.8.2. Molecular Dynamic Simulations

The energy-minimized structures were subjected to MD simulation in a periodic boundary condition using the Desmond [64] (Schrödinger’s maestro, Desmond, NY, USA) software package with the OPLS3e force field. The systems were solvated with multiple layers of water molecules using the TIP3P water model. Further, the system was ionized and neutralized with 0.15 M of NaCl. The temperature and pressure were maintained using a Noose–Hover thermostat and Martyna–Tobias–Klein barostat respectively using a timestep of 2 fs. The cut-off distance to coulombic interaction was set to 9.0 Å. The monomers were simulated at various temperatures ranging from 300 K to 450 K for 50 ns. A choice of the temperature and time threshold was created to produce a partially unfolded protein structure. To study protein aggregation, the evolved protein structure was docked using the PIPER (Schrödinger, maestro) [65,66,67], where ten thousand conformations were generated, and the output was ranked according to the size of the cluster. The docked structures with the largest cluster size were selected. Structures were energy minimized and followed by a simulation of 200 ns at room temperature (300 K).

## 5. Conclusions

In the present study, by performing Raman spectroscopic investigations and Molecular Dynamic simulations, we obtained insights into the multiple physiochemical changes in the protein backbone, the amino acids, and other inter-/intra- molecular bonds of HEWL at alkaline pH. In brief, the environment surrounding the Tryptophan residues, analyzed from the I_1341_/I_1363_, provided us insight into the conformation of the protein, i.e., whether it was unfolded or folded. Analyzing various backbone peaks such as those at 900 cm^−1^, 932 cm^−1^, and 1446 cm^−1^ showed us that the protein was also going through multiple conformations accompanied by a decreasing α-helical content. The peaks of the amino acid residues such as those at 759 cm^−1^, 876 cm^−1^, and 830 cm^−1^ demonstrate the environmental changes experienced by these residues, thus pointing to their positions during and after aggregation. Raman spectroscopy and MD simulations suggested that the aggregation process in pH 12.2 (ambient temperature) was initiated by breakage of the disulphide bond. This led to the destabilization of the α-helices and the -SH in the cysteines were available for intermolecular bond formations as the intramolecular distances increased vastly. This facilitated β-sheet formation in the aggregated protein, as seen in the Raman analysis of the Amide I band, showing a decrease in α-helices and an increase in β-sheets. While each of the different techniques used in earlier reports provides certain aspects of the protein aggregation process, Raman spectroscopy with MD simulations provides a much-detailed microscopic picture of the aggregation process. Thus, Raman spectroscopy combined with MD simulations proves to be an effective tool to study the protein aggregation process.

## Figures and Tables

**Figure 1 molecules-27-07122-f001:**
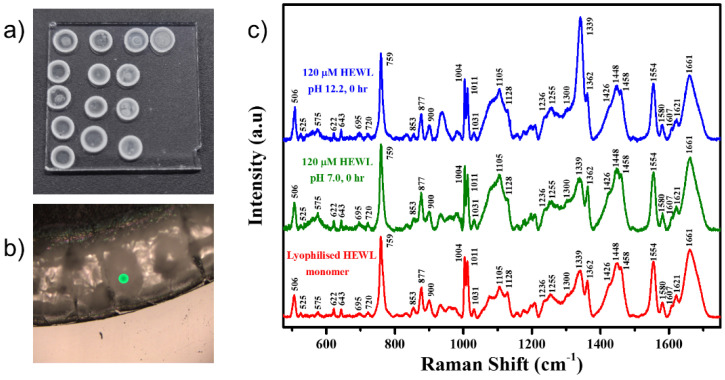
(**a**) Optical image of the rings formed when the HEWL sample was drop-casted onto a hydrophobic glass substrate and dried in a desiccator; (**b**) 10× magnified image of the outer ring of the drop, the site from where the Raman spectrum was collected; (**c**) Raman spectra collected from the lyophilized HEWL monomer (red), rings of the 120 μM HEWL in pH 7.0 (green), and pH 12.2 (blue) at 0 h of the experiment.

**Figure 2 molecules-27-07122-f002:**
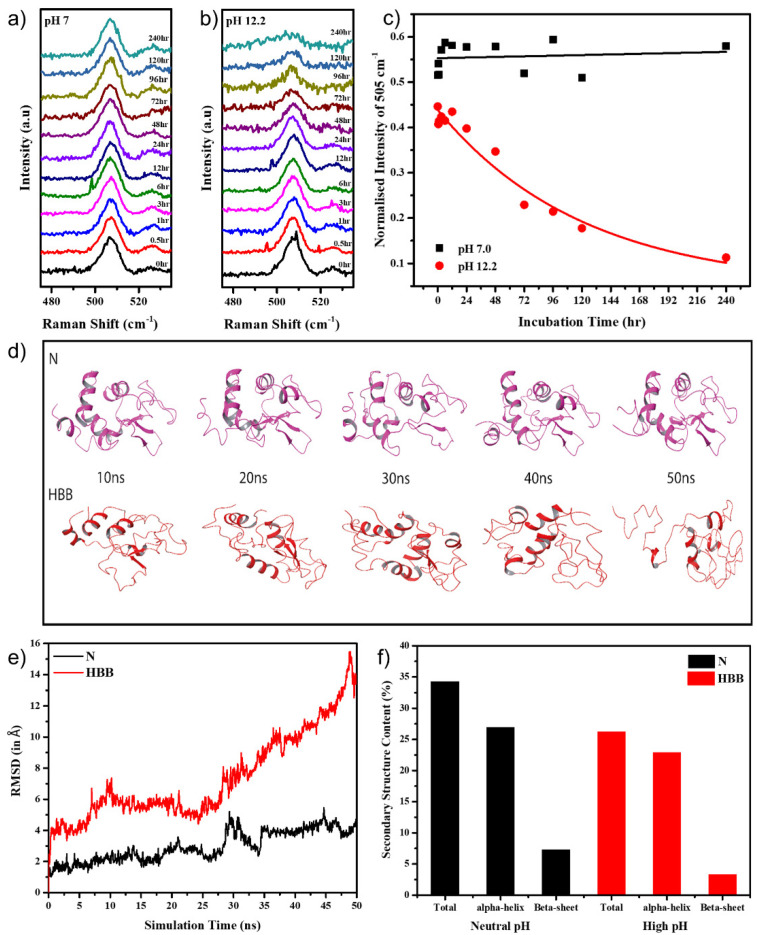
Intensity of 505 cm^−1^ Raman peak in (**a**) pH 7.0 and (**b**) pH 12.2; (**c**) intensity vs. time plot for 505 cm^−1^ Raman peak; (**d**) various frames of the HEWL monomer at different simulation time points, in N and HBB conditions; (**e**) RMSD of the HEWL monomer in neutral pH and high pH with disulphide bonds broken in a 50 ns simulation; (**f**) secondary structure content in N and HBB after the simulation.

**Figure 3 molecules-27-07122-f003:**
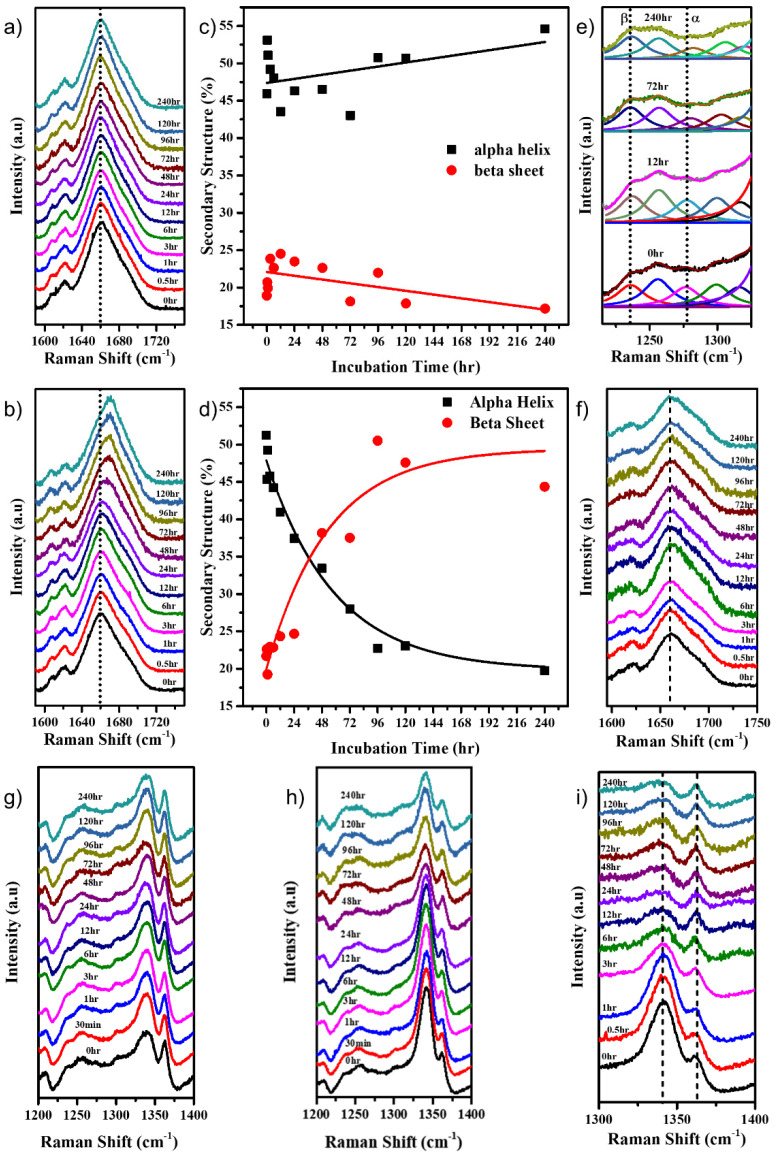
Stack plot of the Amide I Raman band of HEWL in (**a**) pH 7.0 and (**b**) pH 12.2; % α-helical and β-sheet content of the HEWL protein incubated in (**c**) pH 7.0 and (**d**) pH 12.2; (**e**) stack plot with Lorentzian fitting of the Amide III Raman band at selected time points for HEWL incubated in pH 12.2; (**f**) stack plot of Amide I Raman band of HEWL incubated in pH 12.2, with iodoacetamide added to it. Stack plot of the 1341 cm^−1^ and 1363 cm^−1^ doublet peaks of HEWL in (**g**) pH 7.0, (**h**) pH 12.2, and (**i**) pH 12.2 with iodoacetamide.

**Figure 4 molecules-27-07122-f004:**
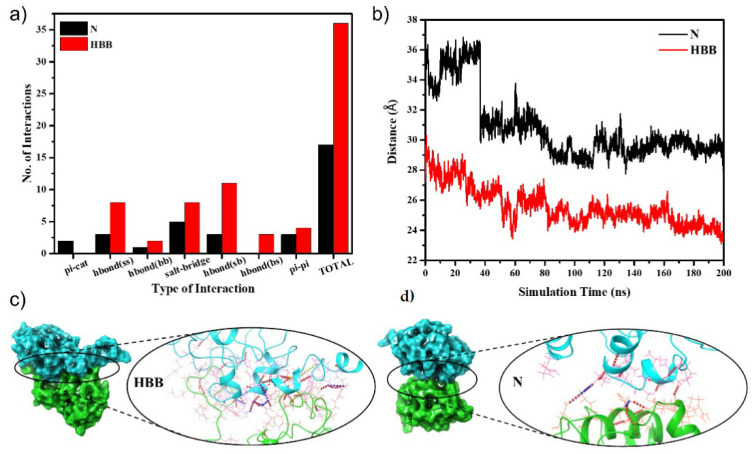
(**a**) Various kinds of interactions amidst the monomers N and HBB; (**b**) binding free energy of the docked proteins with the monomers N and HBB; (**c**) the total number of interactions in the dimers of the N and HBB systems; (**d**) a visual representation of the docked protein dimers in N and HBB indicating a more significant number of interactions in HBB as compared with N.

## Data Availability

Not applicable.

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
