# Peer review of "Insights on Aggregation of Hen Egg-White Lysozyme from Raman Spectroscopy and MD Simulations"

_molecules, 2022, doi:10.3390/molecules27207122_

Round 1
Reviewer 1 Report
The scientific content of the ms. describes the Raman spectroscopic investigations and Molecular Dynamic simulations study in order to obtain insights into the multiple physiochemical changes in the Hen Egg-White Lysozyme (HEWL) protein backbone, the amino acids, and other inter-/intra- molecular bonds of HEWL at alkaline pH. The introduction highlights the state of the art and the novelty of the work well-enough. The present study is well-presented and the experimental findings are discussed satisfactorily. However, there are some points that should be considered in order this ms. to be accepted for publication:
(a) Results, 2.2 Protein Interactions and Aggregation: The authors mention in the “Introduction” that to facilitate aggregation in the acidic conditions we need to provide external stimuli like increased temperature or agitation. As shown in Fig. 3a the position of the Amide I band of the protein, which provides information about the secondary structure of the protein, at neutral pH is constant indicating the stability of the HEWL structure. This is not the case, however for alkaline conditions, where a structure unfolding is noticed, which is consistent with previous studies. Therefore, afterwards the authors, investigate the impact of iodoacetamide, an inhibitory factor for the disulfide bonds formation under pH=12.2 to see if the aggregation is prevented. What would it happen if we add this factor under neutral pH? This would be beneficial to verify the stability of the structure under neutral pH?
(b) Results, 2.2 Protein Interactions and Aggregation: page 7, lines 250-253. Indicate that you are referring to pH=12.2.
(c) Results, 2.2 Protein Interactions and Aggregation: page 8, lines 259-265. “Iodoacetamide…its stability”. This information should be transferred earlier in the 2.2 part, when the addition of iodoacetamide is mentioned for the first time.
(d) Results, General Observation: Important figures are missing from the main ms. being illustrated in the Supporting Information. For instance, Fig. S5 and Fig. B should be transferred in the main ms.
(e) Materials and Methods, 4.4 Raman spectroscopy: The authors performed the Raman measurements by exciting with a 532 nm laser. Spectral profile strongly depends on the wavenumber of the laser excitation in some cases for proteins due to the characteristic optical response of the spectrometers and the detector used, or Resonance Raman effect. Did the authors try another laser line? Do they know if any Resonance effect contribute on the measurements?
(f) Supporting Information: The authors should be consistent on how they present Figures. Fig. A and Fig. B is not an appropriate way based on the others Fig. S1, S2, etc. of Supporting Information.
(g) Supporting Information: Fig. S3a could be exploited as Graphical Abstract.
Author Response
Dear Referee,
We are very thankful for your valuable comments and have worked on the manuscript trying to include your various suggestions. Please find attached the file with the responses for your kind comments.

Reviewer 2 Report
The research article, 'Insights on Aggregation of Hen Egg-White Lysozyme from Raman Spectroscopy and MD Simulations' by Chalapathi et al. is a good piece of work on the spontaneous aggregation of hen egg-white lysozyme (HEWL) in alkaline pH 12.2 at ambient temperature. I would recomend the article for publication in Molecules after minor revision. Here are my comments that need to be addressed.
1. Collect atleast one Raman spectra in acidic pH to show the contrast between pH12 and acidic pH.
2. Explain the decreasing Raman intensity at 506 cm-1 with incresing aggregation.
3. Does the MD simulation or Raman spectra predict formation of polymer of the HEWL beyond aggregation?
you can follow and cite the following papers
https://doi.org/10.1039/C3CP54247E
https://doi.org/10.1002/jrs.5567
4. I would suggest to collect the Zeta of HEWL before and after formation of aggregates to address the change in zeta potentials.
5. Provide the chemical structure and MD interaction of iodoacetamide with the HEWL in the supporting information, if possible.
Author Response
Dear Referee,
We would like to thank you very much for your kind suggestions and valuable comments. Please find attached a file with all the responses for your comments ,and we have tried our best to incorporate the appropriate ones in the manuscript according to your suggestions.
